# Experimental Study on the Mechanical Behavior of Orthodontic Arches Exposed to the Environment in the Oral Cavity

**DOI:** 10.3390/children9010107

**Published:** 2022-01-14

**Authors:** Alexandru Stefan Zalana, Maria Dămășaru, Edgar Moraru, Ciprian Ion Rizescu, Simina Neagoe (Chelărescu), Mariana Păcurar

**Affiliations:** 1Drd Orthodontic Department, Faculty of Dentistry, University of Medicine, Science and Technology G. E Palade Tg Mures, 540142 Targu Mures, Romania; alexandruzalana@gmail.com (A.S.Z.); damasaru.maria@gmail.com (M.D.); 2Department of Mechatronics and Precision Mechanics, University Politehnica of Bucharest, 060042 București, Romania; eddy_milan91@yahoo.com; 3Periodontological Department, Titu Maiorescu University, 540139 Targu Mures, Romania; chelarescu.simina@yahoo.com; 4Pedodontic Department, Faculty of Dentistry, University of Medicine, Science and Technology G. E Palade Tg Mures, 540142 Targu Mures, Romania; marianapac@yahoo.com

**Keywords:** orthodontic arches, dentistry, wires, oral cavity environment, deformations, orthodontic treatment, orthodontics

## Abstract

Background. The arches used in orthodontic therapy are subject to increasing physical and chemical stresses. Purpose of the study: This in vitro experimental study aims to highlight and compare the main mechanical properties of orthodontic arches. Materials and Methods: We used 40 springs, 2 materials, 20 of Ni-Cr and 20 of Co-Cr, of different diameters, 0.7 mm 0.8 mm and 1.2 mm, subjected to the environment of artificial saliva and artificial saliva with cola for one month and two months, respectively. Five springs of each material were tested at different times: T0, before application in the oral cavity, then at time T1, T2, T3, T4. Three lengths of the lever arm were considered for bending forces acting on the springs (dental wires). These lengths were 15, 10 and 5 mm. The wires were tested under the action of bending forces on a Hans Schmidt HV 500N stand, obtaining the characteristics of the wires: deformation-force-time. Results: Graphical determinations show that the degree of deformation of the wires is influenced by the applied force, diameter and obviously by the immersion time, respectively by the type of solution in which the springs were immersed. Conclusions: The final degree of bending is higher for Co-Cr arcs than for Ni-Cr at all three dimensions.

## 1. Introduction

Orthodontic mobile appliances are made of an acrylic base, active and anchoring elements made of orthodontic wires. The arches used in mobile or fixed orthodontic therapy are subject to physical and chemical stresses that sum up. The deformations printed by the doctor during the activation of the devices overlap with the deformations produced by the patients during the handling of the mobile orthodontic appliances, and to these is added the chemical degradation due to the oral environment (saliva with microorganisms, respectively ingested food and liquids).

Orthodontic biomechanics [1] is based on the principle of storing elastic energy and converting it into mechanical energy during tooth movement. Optimal control of tooth movement requires the application of a system of special forces, through accessory elements such as dental springs/auxiliary springs.

An element of an orthodontic arch can be subjected to three types of forces: traction, bending and shear (torsional) force [2]. The tensile force causes an elongation in the direction of the application of the force, the bending force determines a contraction in the direction of application, and the shear force produces a sliding displacement of a portion or torsion [3]. All three types of force are present in the case of any mechanical test and may vary in intensity depending on the different areas within the structure of that element [4]. While testing by applying these types of forces reproduces quite well the conditions to which the spring elements are subjected during clinical use, the response of internal structures can be much more complex [4].

In the case of orthodontic therapy, of particular interest is the biocompatibility of the main force-triggering elements, namely orthodontic arches [5]. Although the permanent improvement of the composition and quality of spring alloys has led to a significant increase in their stability in the oral environment and implicitly their biocompatibility, the phenomena of matting and corrosion characteristic of metals are still present [6], especially since the devices are kept in the oral cavity for quite a long time (2–3 years).

There are studies that show the damage of orthodontic appliances and arches due to the environment of the oral cavity [7] such as the action of fluoride in toothpaste [8] on orthodontic appliances and arches. It is known that fluoride [9] affects more the arches which contain NiTi and less the SS and CoCr [10], so we tried to expose for a longer period these type of arches that are used for longer periods of time in removable appliances and lingual or practitioner constructed arches [11].

Taking into account the long duration of orthodontic treatment, we consider relevant the immersion test of orthodontic arches for 30 days, respectively 60 days in a mixture of artificial saliva and cola, thus simulating the consumption of sweet (about 2 litre of cola consumption daily), carbonated drinks for about 2 years of orthodontic treatment.

## 2. Materials and Methods

### 2.1. Purpose of the Study

This in vitro experimental study aims to highlight and compare the main mechanical properties (tensile strength and bending/bending strength) of orthodontic arches of Ni-Cr and Co-Cr of different diameters immersed in solutions similar to those in the oral cavity, artificial saliva and artificial saliva combined with cola, at different intervals of time.

### 2.2. Working Method

We used Ni-Cr and Co-Cr wires, of different diameters, 0.7 mm, 0.9 mm and 1.2 mm, subjected to the oral environment of artificial saliva and artificial saliva combined with cola for one month, respectively two months. Springs were purchased from the same manufacturer, with the technical characteristics of provenance (Figure 1).

For each thickness of wire and material three samples were considered, as an example, 3 pieces (samples) of wire with a length of 50 mm and a thickness of 0.7 mm of Ni-Cr. These three samples for each case were tested in a laboratory. The results were quite equal, with no significant differences. Then, they were measured in several ways. After performing the measurements, they were immersed in artificial saliva (Fusayama Meyer) and in a solution of artificial saliva mixed with cola in a certain fixed percentage. Orthodontic arches were kept in an incubator with a constant temperature similar to the temperature of the human body. During this time, the mixtures of solutions were constantly refreshed at an interval of 2 days. At 30 and 60 days, respectively, the set of measurements was resumed.

The experimental results are based on three determinations on each case. If one of the three values was 10% different from the other two, the test was repeated. In conclusion, there was no variation greater than 10% of the three determinations. For the graphical representation of the results, the Matlab 2009a program from the Polytechnic University of Bucharest was used. It was established that the arrow of orthodontic arches can vary between 0 and 10 mm and the evolution of the deformation forces according to the arrow was followed. In addition, for all the tests performed, the evolutions in time of the bending forces were recorded with the help of the data acquisition program of an Imada transducer.

The springs were embedded in a construction with an acrylate base and rigidly fixed. On a cylinder, which has the role of simulating a tooth, the above-mentioned wires were subjected to bending forces (Figure 2a,b). The diameter of the cylinder was 8 mm.

Subsequently, the springs were tested on a Hans Schmidt stand consisting of a platform table equipped with adjustable clamps and the possibility of moving on the x and y axes by means of micrometric screws (Figure 3).

An IMADA force transducer is operated on the work platform, which measures the pressing force according to the travel length; data are transmitted in real time to a computer connected to this device. The maximum force that the transducer can measure is 500 N. The received data is processed by the computer, which generates a graphical interpretation (Figure 4a,b).

For the first measurements we used 40 springs, 20 of Ni Cr and 20 of Co Cr, of different diameters, 0.7 mm; 0.9 mm and 1.2 mm, subjected to the environment of artificial saliva and artificial saliva with cola for one month and two months, respectively.

Springs of each material were tested at different times: T0, before application in the oral cavity (new springs); T1, 1 month after immersion in artificial saliva; T2, 2 months after immersion in saliva artificial, T3, 1 month after immersion in artificial saliva + coca cola; T4, 2 months after immersion in artificial saliva + coca cola. Three types of lengths of the lever arm with which the spring was deformed were simulated, 15(l3), 10(l2) and 5(l1) mm, which represent via analogy the distance from the base of the device where the spring is inserted to the contact with the tooth.

These wires were tested under the action of bending forces on a Hans Schmidt HV 500 N stand, equipped with an IMADA force transducer, the data received being received by a computer that generated a graphical interpretation obtaining the characteristics of the wires: deformation-force-time.

The bending deflection (deformation) was measured for the three lengths of the springs at the two different alloys (Figure 5) using a Mitutoyo distance measurement system. The orthodontic wires were considered recessed springs at one end, actuated with a force at three different lengths.

Three measurements were performed for each spring and material thickness at 3 different arm lengths. For each length, for the three measurements the average value was computed, and the result obtained was entered in the work tables.

The method that is currently used is linear regression with the determination of the regression factor R^2^ [13]. The equation of the line that approximates the obtained points is also displayed. We expected a linear behavior of these wires and the results sustained our expectation [14]. Figure 6 shows the graph of linear regression for the 0.7 mm diameter NiCr wire.

## 3. Results

For wire of 0.7 mm diameter Ni-Cr, bent at an arm of 5, 10 and 15 mm, the corresponding deformations (in mm) are listed in Table 1.

As can be seen in Table 1, as the spring arm increases, the deformation in mm decreases, so the longer springs trigger a smaller displacement of the tooth. Comparing the deformations of the springs over time (final degree of bending), we found a linear increase from T0, 1 month after immersion in saliva, (from 3 to 3.5 mm), respectively from 1 mm, to 1.28. The exception is the springs with a length of 5 mm, which after a month of immersion has a lower degree of deformation compared to T0.

At moments T1 and T2, respectively at immersion in artificial saliva and cola, at 1 month/2 months there is a decrease in the final degree of bending for all three lengths of the spring arm (Figure 7a–c).

The influence of the attack is noticeable. The deformation characteristic is no longer linear, so obviously the springs deform a lot in the environment of the oral cavity (humid environment), to which are added the acidic components of different foods/drinks (coca cola).

For 0.7 mm diameter Co-Cr wire, bent at an arm of 5, 10 and 15 mm, the corresponding deformations (in mm) are listed in Table 2.

As can be seen in Table 2, as the spring arm increases, the deformation in mm increases, one month after immersion in artificial saliva. Thus, the longer springs in the Cr-Co trigger a greater displacement of the tooth.

Comparing the deformations of the springs over time (final degree of bending), we found a linear increase from T0, 1 month after immersion in saliva, (from 2.7 mm to 3.20), respectively from 1.2 mm to 2.10. Springs of 5 mm length, after 2 months of immersion in artificial saliva, have a lower degree of deformation compared to T0. When immersed in artificial saliva, they have a linear increase, compared to the spring springs of 10 and 15 mm, respectively. This evolution is shown in Figure 8a–c.

For the 0.9 mm Ni-Cr wire, the results are listed in Table 3.

As can be seen in Table 3, as the spring arm decreases, the deformation in mm decreases, 1 month after immersion in artificial saliva and cola, respectively 2 months after immersion.

Exceptions are these springs at an interval of 1 month of immersion in artificial saliva, when we found a slight increase in the degree of deformation from 0.87 mm, as it was initially, to 0.97 mm. Longer Ni-Cr springs with a diameter of 0.9 mm trigger a smaller displacement of the tooth, not being indicated in the displacement stages, only in the leveling phases. This evolution of the deformations of Ni-Cr arcs is shown in Figure 9a–c.

For the 0.9 mm Cr Co wire, the results are listed in Table 4.

As can be seen in Table 4, the deformation of wires with a larger diameter, 0.9 mm, is more obvious in mm increases at 1 month after immersion in artificial saliva and cola and respectively 2 months after immersion for all three dimensions of arm length. Exceptions are the springs with 5 mm arm at an interval of 2 months of immersion in artificial saliva, when we found a slight decrease in the degree of deformation from 0.49 mm to 0.26 mm. This evolution of the deformations of Co-Cr arcs is shown in Figure 10a–c.

For the 1.2 mm Ni-Cr wire, the changes are shown in Table 5.

As can be seen in Table 5, the deformation of the wires with larger diameter, 1.2 mm, is more obvious, with a different evolution from the rest of the wires. Thus, for the arm length of 15 mm, the deformation increases very little from T0 to T1, from 0.46 to 0.48 at 1 month after immersion in artificial saliva and cola, respectively to 0.59 at 2 months after immersion in artificial saliva and respectively to 0.82 for immersion in artificial saliva and cola at 2 months.

For the 10 mm arm length, the deformation increases from 0.32 (T0) to 0.37 (T1) and 0.40 (T2). Exceptions are 1.2 mm diameter wires with a 10 mm arm 2 months after immersion in artificial saliva when the deformation decreases very little, from 0.32 to 0.14. The diagram of these changes is shown in Figure 11a–c.

For the 1.2 mm diameter Cr-Co wire, the changes are shown in Table 6.

As can be seen in Table 6, the deformation of Cr-Co wires with a diameter greater than 1.2 mm is more obvious, with a different evolution from Ni-Cr wires. Thus, for the arm length of 15 mm, the deformation decreases for T0 to T1, from 3.44 to 1.45 at 1 month, respectively 2 months after immersion in artificial saliva and immersion in artificial saliva with cola.

The evolution of wires with the 10 mm arm is not rectilinear, in the sense that it decreases at T1 from 0.65 to 0.33, then increases to 0.86 at T1.

The evolution of 5 mm long wires has a decreasing tendency, except for the T2 moment, when the wires were kept in artificial saliva for 2 months.

The evolution of these springs is represented in Figure 12a–c.

The bending force varies between 10 and 30 N, which can be useful for the design of dental appliances. These tests were used to determine the stiffness of an orthodontic spring and will help the doctor to decide on the diameter of the wire to use depending on the type of tooth movement during orthodontic treatment. The stiffness, k, of a body is a measure of the resistance offered by an elastic body to deformation. In this case the stiffness of the orthodontic arcs was determined, considering the elastic bodies with respect to Hooke’s law.

The dynamics of the stiffness of tested wires considering different diameters is shown in Figure 13, Figure 14 and Figure 15. The stiffness of the springs increases with the increase of the diameters of the springs.

## 4. Discussion

The experimental method of measuring the mechanical behavior of springs is often the same, both in terms of tensile and bending tests. For each diameter size and each wire material three experimental set-ups were developed. The orthodontic wires were considered recessed springs at one end, actuated with a force at three different lengths. For each diameter size and material the average value was computed. In the present study, we observed a greater impairment of the arcs immersed in the fluorinated solution, compared to those immersed in cola. Thus, it was possible to highlight a greater surface area and depth of the cracks on the surface of the springs. Kusy and Dilley [15] consider the three-point bending test to be more accurate than the four-point, with smaller variances. The same authors highlighted the possible errors produced as a result of the spring slipping from the supports, due to excessive bends [15]. To ensure that a spring is tested within its mechanical properties [16], Kusy recommends that the deformation not exceed 5% of the length of the segment, which means that in the case of a 15 mm segment, the maximum deformation can be 0.75 mm, or in the oral cavity springs suffer much larger deformations between 2–4 mm.

The studies of Abalos [17], which quantify the hydrogen uptake of orthodontic springs via short-term immersion in hydrofluoric acid solution, revealed increased amounts of hydrogen on the surface of all springs with consequent degradation of the mechanical performance of immersed springs in comparison with the new ones. The same study [17] found important changes in the surface topography of submerged arcs with increasing corrosion roughness of NiTi arcs, the appearance of uniform corrosion marks of β-Ti and signs of corrosion unevenly distributed on the surface of the springs.

The characterization of different springs in terms of their mechanical properties is an extremely important step in understanding their clinical behavior. Accurate knowledge of the advantages and disadvantages of a specific alloy helps the clinician to select the perfect arch from the multitude of products on the orthodontic market.

Parvizi’s study [18], performed in a set-up similar to ours, did not detect a significant increase in the forces of Ni-Cr arcs, when deformations of 2 mm versus 4 mm occurred; however, when the tests were performed on a three-dimensional model, the forces were significantly increased at all springs. The correlation of the stiffness of springs of the same size but from different alloys reveals the practical importance of knowing the forces developed by springs. For example, while a 0.016 × 0.022 Ni-Ti arc develops a force of 3.35 (0.17) N, an SS arc develops a force of 5.07 (0.42) and a β-Ti arc of 3.52 (0.81) N. Thus, the tension transmitted at the level of the periodontal ligaments in orthodontic biomechanics varies from low values to high ones depending on the bending characteristics of the material used. 

From an orthodontic point of view, we are more interested in the forces that appear at 1 mm and 2 mm displacements of the wires, when forces are between 2–4 N.

Increasing the diameter by only 0.2 mm (from 0.7 to 0.9) the deformation of the wire decreases by half for wires exposed to chemical attack. 

The deformation decreases considerably as the diameter increases (from 0.7 to 0.9 mm), due to the increase of the wire rigidity, which increases approximately 2 times in the case of CoCr, over 2.5 times in the case of NiCr (T0), over 2.5 times in the case of NiCr and CoCr (1 month), 2 times in the case of NiCr and almost 3 times in the case of CoCr (2 months)—Figure 12.

In most cases it was observed that the deformation forces decrease significantly after immersion in the experimental fluids and the immersion time also affected the deformation forces (on 2 months wires the deformation forces are smaller than the ones applied to obtain the same deformation on T0 wires, so they are easier to be deformed after immersion).

The influence of Coca-Cola is not major, even in cases where the wires showed greater deformation forces for the models immersed in saliva + cola, than those that were immersed only in saliva.

In order to obtain the deformation of 2 mm (arm 5 mm) for the CoCr wire (diameter 1.2 mm) we needed a force of 61 N. After a month in saliva the same deformity was already obtained at 36 N and in 2 months at 22 N force, and in the case of Coca-Cola at 37 N and 23 N, respectively. Thus, it is obvious that after immersion in experimental fluids the material underwent changes in terms of mechanical properties and became easier to deform (immersion time as well influences this). In contrast, Coca-Cola does not significantly change its properties compared to cola-free saliva (even sometimes with cola they were more resistant). The stiffness of the orthodontic arc increases with the increase of the diameters of the springs. Our tests are useful for orthodontists to determine the diameter of the wire to be used. High stiffness of the wire will lead to high forces, which may exceed the values recommended in orthodontic treatment.

## 5. Conclusions

Springs (wires) with a diameter of 0.7 mm Ni-Cr and Co-Cr show a slight decrease in deformation force in proportion to the exposure medium and the duration of exposure.

Co-Cr wires of 0.9 mm have an approximately unchanged strength regardless of the exposure environment and the duration of exposure. The strength of Co-Cr 0.9 mm is higher than that of Ni Cr 0.9 mm.

Ni-Cr wires of 0.9 mm show a decrease in strength, but the decrease is indifferent to the exposure environment and decreases over an increased duration of exposure.

Co-Cr wires of 0.9 mm do not show a decrease in strength at 1 month after exposure, but show a marked decrease after 2 months of exposure regardless of the exposure medium, and the strength of Co-Cr 0.9 mm is slightly larger than 0.9 mm Ni-Cr.

Ni Cr wires of 1.2 mm do not show change for the resistance force in proportion to the exposure medium and the duration of exposure.

Co-Cr wires of 1.2 mm show a slight decrease in strength in proportion to the exposure medium and the duration of exposure.

The larger the diameter of the orthodontic arches, the less they are mechanically affected by exposure to the oral cavity.

The final residual deflection of the spring is higher for Co-Cr arcs than for Ni-Cr considering all three arm lengths l_1_, l_2_ and l_3_.

## Figures and Tables

**Figure 1 children-09-00107-f001:**
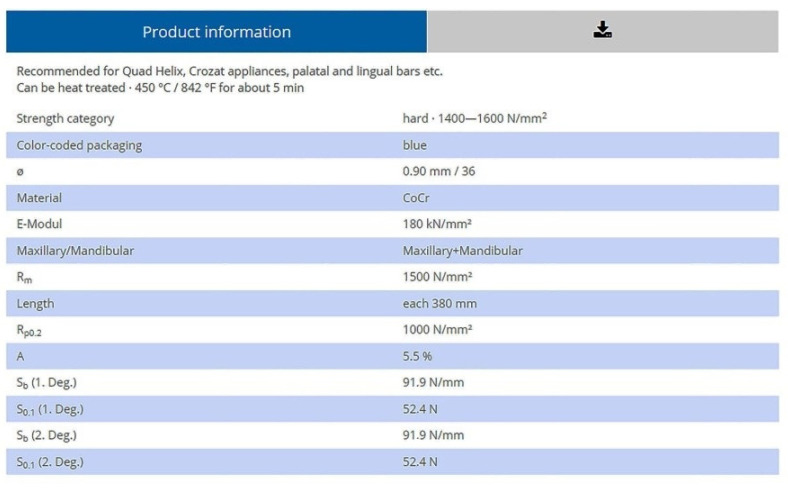
Technical characteristics of Co-Cr springs [12].

**Figure 2 children-09-00107-f002:**
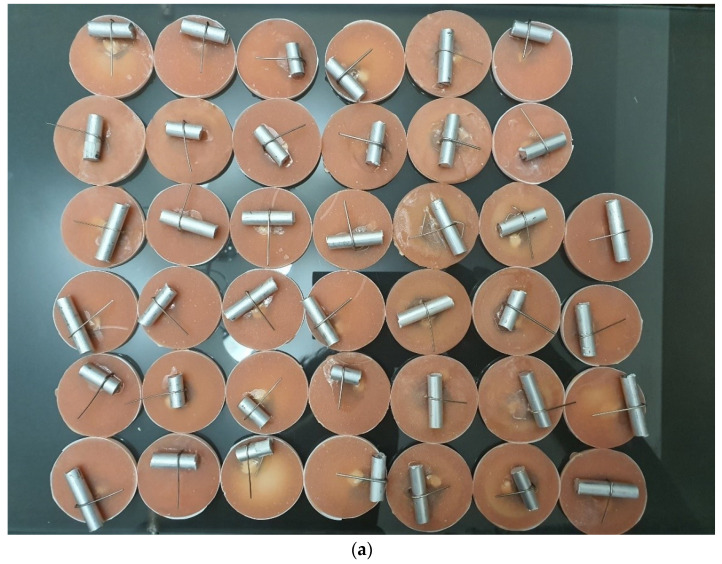
(**a**) Cylinders with bent springs; (**b**) wire distance measurement.

**Figure 3 children-09-00107-f003:**
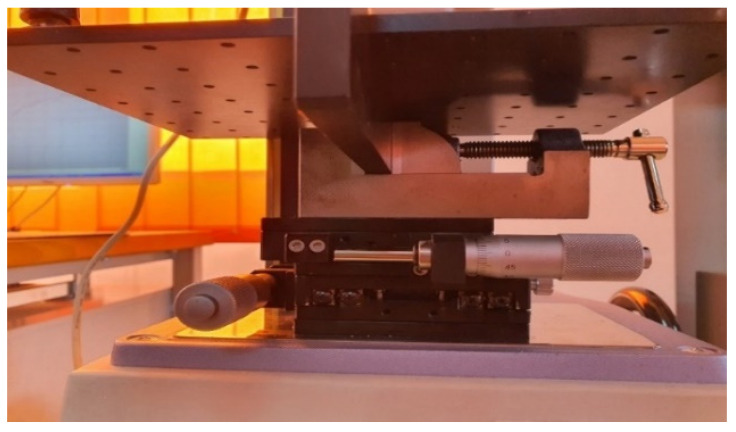
XY table from HV 500 set-up.

**Figure 4 children-09-00107-f004:**
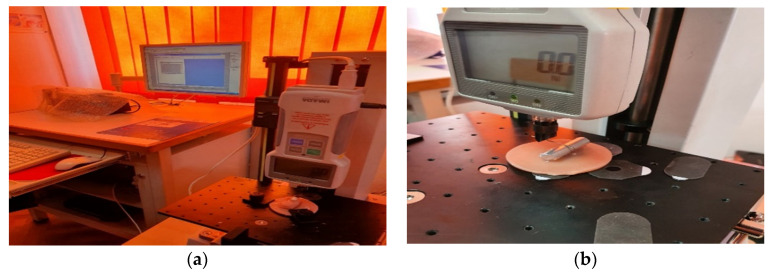
(**a**) HV 500N stand used for spring testing; (**b**) force transducer.

**Figure 5 children-09-00107-f005:**
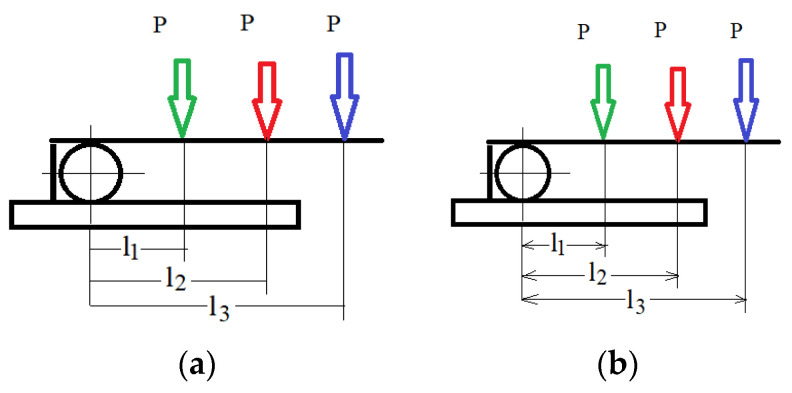
(**a**) Deflection of Ni-Cr arcs; (**b**) deflection of Co-Cr.

**Figure 6 children-09-00107-f006:**
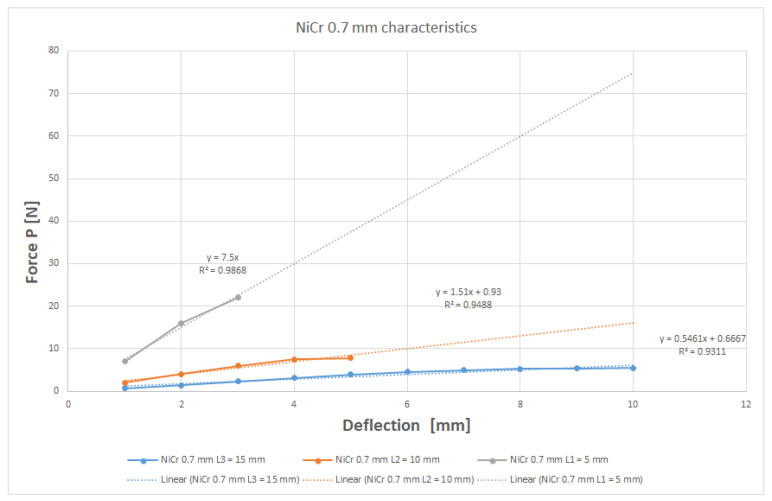
Graph that shows the linear regression for NiCr 0.7 mm.

**Figure 7 children-09-00107-f007:**
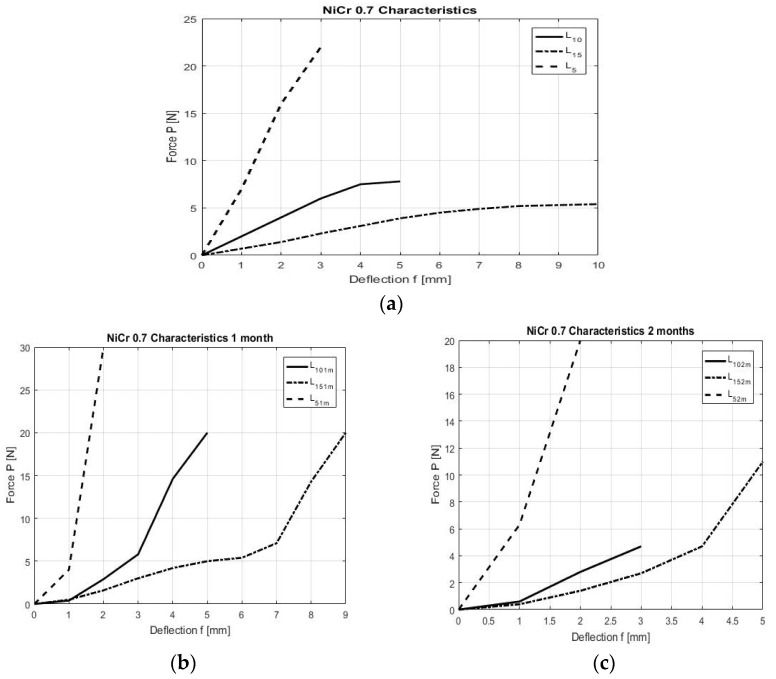
(**a**) Ni-Cr bending with a diameter of 0.7 mm without attack. A linear behavior is observed for each length of the arc. (**b**) NiCr bending with saliva attack, 1 month; (**c**) NiCr bending with saliva attack, 2 months.

**Figure 8 children-09-00107-f008:**
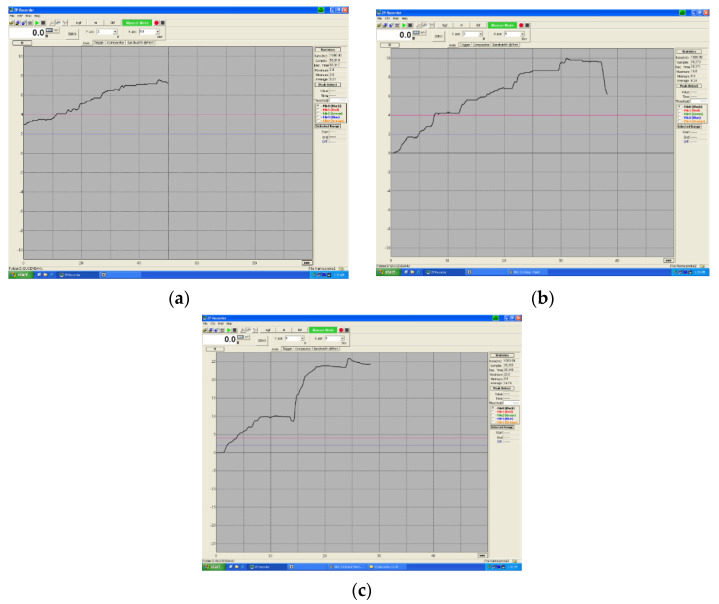
(**a**–**c**) Evolution of Cr-Co spring deformations (time-force).

**Figure 9 children-09-00107-f009:**
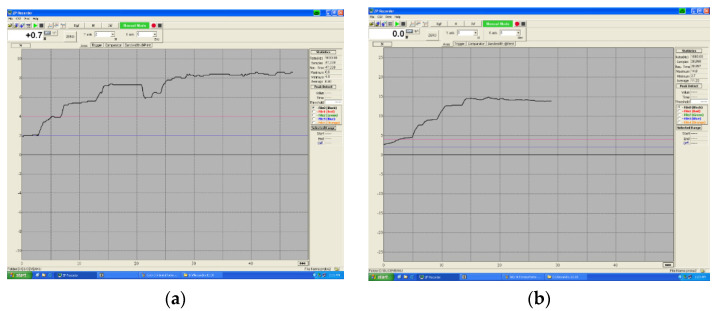
(**a**–**c**) Evolution of Ni-Cr arc deformations, diameter 0.9 mm.

**Figure 10 children-09-00107-f010:**
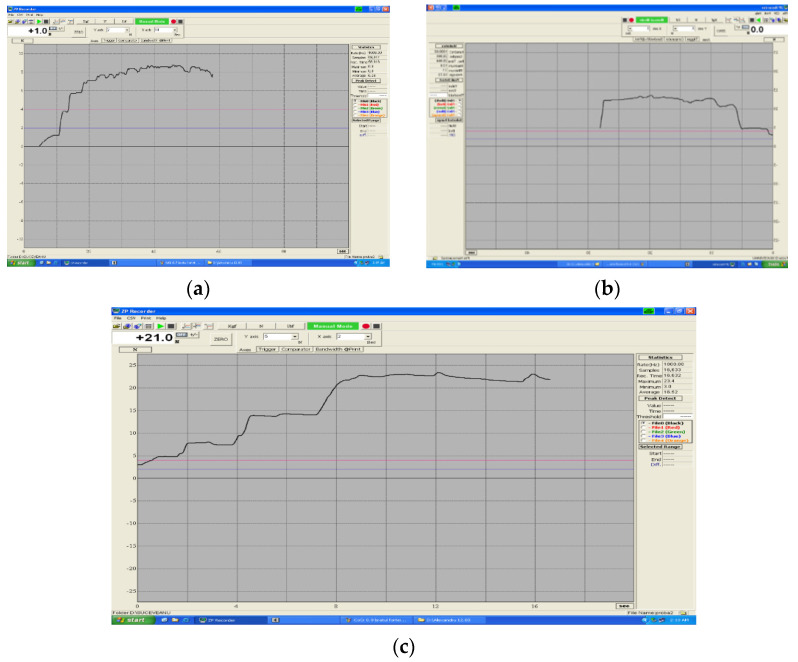
(**a**–**c**) Evolution of Cr-Co spring forces, diameter 0.9 mm.

**Figure 11 children-09-00107-f011:**
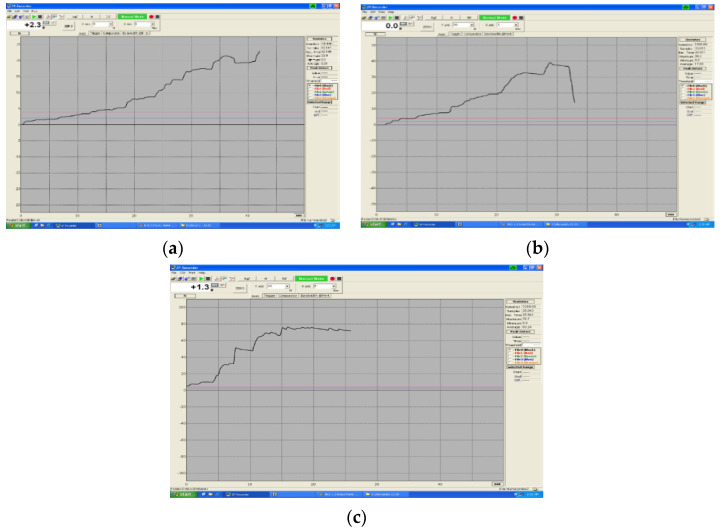
(**a**–**c**) Evolution of Ni-Cr arc forces, wire diameter 1.2 mm.

**Figure 12 children-09-00107-f012:**
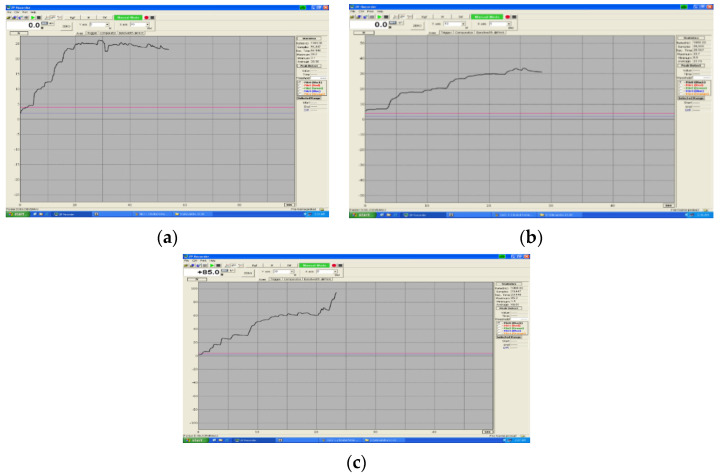
(**a**–**c**) Evolution of Co-Cr arc forces, diameter 1.2 mm.

**Figure 13 children-09-00107-f013:**
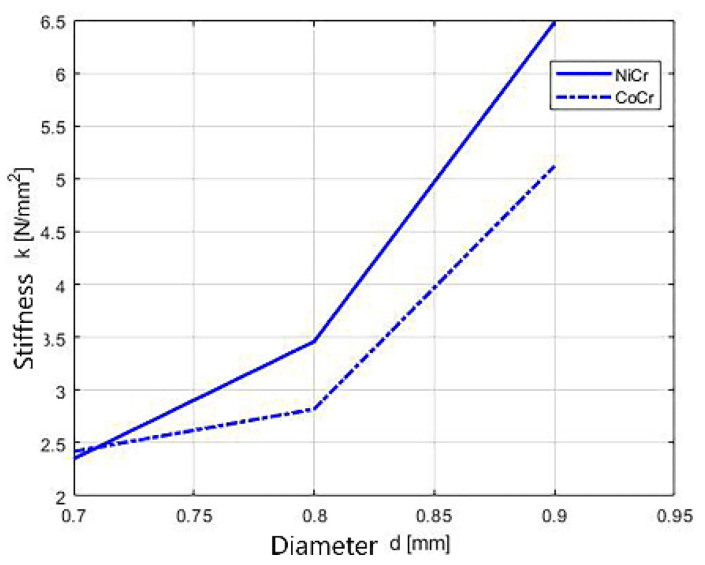
Stiffness dynamics of orthodontic wires.

**Figure 14 children-09-00107-f014:**
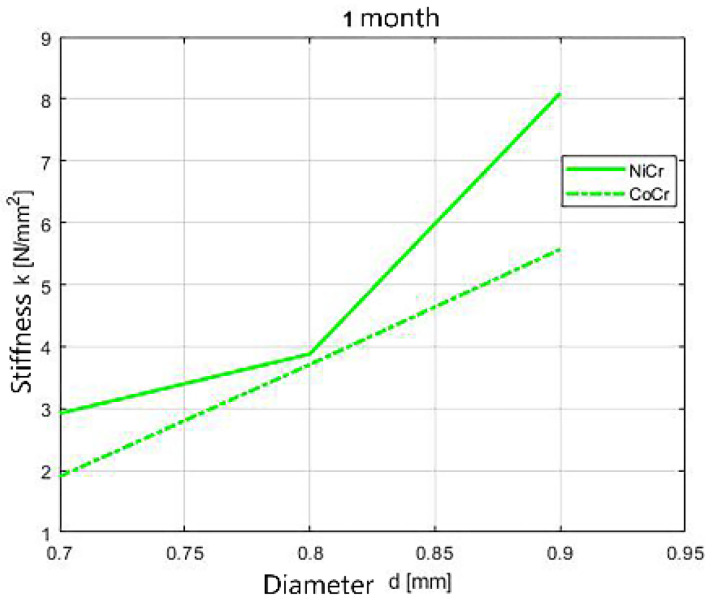
Stiffness dynamics of orthodontic wires, 1 month immersion.

**Figure 15 children-09-00107-f015:**
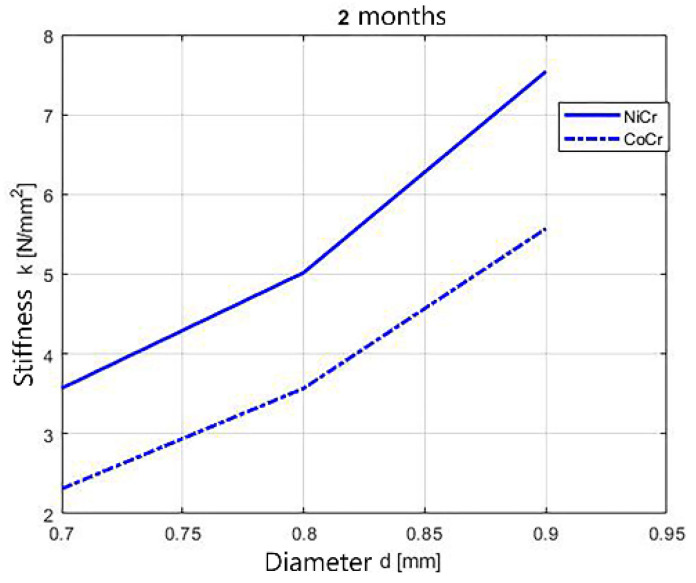
Stiffness dynamics of orthodontic wires, 2 months immersion.

**Table 1 children-09-00107-t001:** Results for Ni-Cr 0,7 mm arches.

Deflection mm	Ni Cr 0.7 mm			Ni Cr 0.7 mm			Ni Cr 0.7 mm			Ni Cr 0.7 mm			Ni Cr 0.7 mm		
				1 Month in Artificial Saliva			2 Months in Artificial Saliva			1 Month in Artificial Saliva + Cola			2 Months in Artificial Saliva + Cola		
	l_3_ 15 mm F(N)	l_2_ 10 mm F(N)	l_1_ 5 mm F(N)	l_3_ 15 mm F(N)	l_2_ 10 mm F(N)	l_1_ 5 mm F(N)	l_3_ 15 mm F(N)	l_2_ 10 mm F(N)	l_1_ 5 mm F(N)	l_1_ 15 mm F(N)	l_2_ 10 mm F(N)	l_1_ 5 mm F(N)	l_3_ 15 mm F(N)	l_2_ de 10 mm F(N)	l_1_ 5 mm F(N)
1	0.7	2	7	0.5	0.4	4	0.4	0.6	6.3	0.4	0.7	1.2	0.4	1	4.2
2	1.4	4	16	1.6	2.9	30	1.4	2.8	20	0.8	2.2	6.2	1.3	3.7	12
3	2.3	6	22	3	5.8		2.7	4.7		1.8	5.1	10	2.4	6.6	21
4	3.1	7.5		4.2	14.6		4.7			3	7.2	16.8	3.1	8.5	
5	3.9	7.8		5	20		11			3.8	8.1		3.8	12	
6	4.5			5.4						4.5	14		4.5		
7	4.9			7.1						5			4.8		
8	5.2			14.3						5.1			5		
9	5.3			20						8			6.6		
10	5.4														
residual deflection of spring mm	3	1	1.05	3.5	1.28	0.31	1.05	0.65	0.66	1.11	1.06	0.6	2.3	1.03	0.59

**Table 2 children-09-00107-t002:** Results for Co-Cr 0.7 mm arches.

Deflection mm	Co-Cr 0.7 mm			Co-Cr 0.7 mm			Co-Cr 0.7 mm			Co-Cr 0.7 mm			Co-Cr 0.7 mm		
				1 Month in Artificial Saliva			2 Months in Artificial Saliva			1 Month in Artificial Saliva + Cola			2 Months in Artificial Saliva + Cola		
	l_3_ 15 mm F(N)	l_2_ 10 mm F(N)	l_1_ 5 mm F(N)	l_3_ 15 mm F(N)	l_2_ 10 mm F(N)	l_1_ 5 mm F(N)	l_3_ 15 mm F(N)	l_2_ 10 mm F(N)	l_1_ 5 mm F(N)	l_1_ 15 mm F(N)	l_2_ 10 mm F(N)	l_1_ 5 mm F(N)	l_3_ 15 mm F(N)	l_2_ de 10 mm F(N)	l_1_ 5 mm F(N)
1	0.9	2.4	7	0.4	0.7	1.6	0.9	1.6	1.5	0.4	0.9	2	0.4	0.7	2.2
2	2	5	6	0.9	1.6	7.6	3	3.5	9.2	1.4	3.1	9	1.5	2.4	6.8
3	3	6.5	7	1.6	2.4	13	2.8	12		2.4	5	18	2.5	3.8	15.2
4	3.8	9.6		2.6	2.1	24	3.2	16.4		3.2	9.7		3	4.9	
5	4.1	4.5		3.3	2.3	12	3.5	18.5		3.7	15.2		3.3	5.1	
6	4.2	4.3		3.6	4	9	3.7			3.9			3.6	14.5	
7	4.4	4.2		3.7	2.4	8.9	13.5			4			3.7		
8	4.4	4.1		4.1	3.4										
9	4.4	4.3		4.3	1.4										
10															
residual deflection of spring mm	2.7	1.2	0.67	3.20	2.10	1.20	2.10	1.45	0.50	2.52	1.8	0.7	2.8	1.28	1.24

**Table 3 children-09-00107-t003:** Results for Ni-Cr 0.9 mm arches.

Deflection mm	Ni Cr 0.9 mm			Ni Cr 0.9 mm			Ni Cr 0.9 mm			Ni Cr 0.9 mm			Ni Cr 0.9 mm		
				1 Month in Artificial Saliva			2 Months in Artificial Saliva			1 Month in Artificial Saliva + Cola			2 Months in Artificial Saliva + Cola		
	l_3_ 15 mm F(N)	l_2_ 10 mm F(N)	l_1_ 5 mm F(N)	l_3_ 15 mm F(N)	l_2_ 10 mm F(N)	l_1_ 5 mm F(N)	l_3_ 15 mm F(N)	l_2_ 10 mm F(N)	l_1_ 5 mm F(N)	l_1_ 15 mm F(N)	l_2_ 10 mm F(N)	l_1_ 5 mm F(N)	l_3_ 15 mm F(N)	l_2_ de 10 mm F(N)	l_1_ 5 mm F(N)
1	2	4	18.2	1	1.5	2	1	1.5	1.8	0.9	2.1	4.5	1.1	1.2	1.7
2	4	9	37	1.9	3.7	19	1.9	3.3	6	1.6	4.4	20	2.2	2.4	5.6
3	5.6	12.8	35	2.9	9.4		2.7	5.4	11.5	2.5	12	31	3.4	4.2	19
4	6	14.4		4.4	27		3.6	7.5		3.2	33		6	8.6	26
5	8	14.2		6.5			4.7	20		4.4			8.4	12.6	
6	8.3			10			5.7			9			10.3	15.2	
7	8.4			15.2			7.2						11.3	28	
8							9.7						11.7		
9															
10															
residual deflection of spring mm	0.87	0.9	1.66	0.97	0.45	0.39	0.76	0.68	0.67	0.76	0.52	0.40	0.57	1.18	1.08

**Table 4 children-09-00107-t004:** Results for Co-Cr 0.9 mm arches.

Deflection mm	Co-Cr 0.9 mm			Co-Cr 0.9 mm			Co-Cr 0.9 mm			Co-Cr 0.9 mm			Co Cr 0.9 mm		
				1 Month in Artificial Saliva			2 Months in Artificial Saliva			1 Month in Artificial Saliva + Cola			2 Months in Artificial Saliva + Cola		
	l_3_ 15 mm F(N)	l_2_ 10 mm F(N)	l_1_ 5 mm F(N)	l_3_ 15 mm F(N)	l_2_ 10 mm F(N)	l_1_ 5 mm F(N)	l_3_ 15 mm F(N)	l_2_ 10 mm F(N)	l_1_ 5 mm F(N)	l_1_ 15 mm F(N)	l_2_ 10 mm F(N)	l_1_ 5 mm F(N)	l_3_ 15 mm F(N)	l_2_ de 10 mm F(N)	l_1_ 5 mm F(N)
1	1.2	4.9	14	2.5	5.2	12	1	1.5	3.1	2	4.4	15	0.9	1.2	2
2	3.8	11	22	4.7	10	18	1.6	3.1	12.2	4.9	11.8	24	2.1	5	11
3	5.8	13	19	6.6	13		2.1	7.9		7.5	16		4.3	10	19
4	7.1	12.7	18	7.6	13.2		3.6	20		8.8	16.5		6	13.2	
5	7.9	11	17	8.2	13.2		5.5			9.6			7.1	13.5	
6	8.4	11	20	8.9	13.2		6.5			10.2			7.9	16	
7	8.6	11	21	9	13.2		8.1			10.4			8.5		
8	8.1	11	20	9.3	13.2		31			10.3			8.7		
9															
10															
residual deflection of spring mm	3.37	0.72	0.49	4.00	1.69	1.13	2.15	0.96	0.26	4.71	1.89	0.20	2.18	3.25	0.85

**Table 5 children-09-00107-t005:** Results for Ni-Cr 1.2 mm arches.

Deflection mm	Ni Cr 1.2 mm			Ni Cr 1.2 mm			Ni Cr 1.2 mm			Ni Cr 1.2 mm			Ni Cr 1.2 mm		
				1 Month in Artificial Saliva			2 Months in Artificial Saliva			1 Month in Artificial Saliva + Cola			2 Month in Artificial Saliva + Cola		
	l_3_ 15 mm F(N)	l_2_ 10 mm F(N)	l_1_ 5 mm F(N)	l_3_ 15 mm F(N)	l_2_ 10 mm F(N)	l_1_ 5 mm F(N)	l_3_ 15 mm F(N)	l_2_ 10 mm F(N)	l_1_ 5 mm F(N)	l_1_ 15 mm F(N)	l_2_ 10 mm F(N)	l_1_ 5 mm F(N)	l_3_ 15 mm F(N)	l_2_ de 10 mm F(N)	l_1_ 5 mm F(N)
1	1.7	3.8	10	2.3	2.5	12.2	2.2	6.4	13.6	2.7	2.8	9.2	1.5	3.8	6.3
2	3.4	7.5	49	4.2	7.3	26	5.1	12.5	29	6.8	8.3	20	3.6	7.6	14
3	5.3	19	75	6.4	12	42	8.1	18	61	10	15	31	6	11.3	22
4	8.7	32.5		8	24	38	11.1	23	51	13.2	19.4	41	8.1	15	
5	14	37		9.7	39		14	36		16	23.5		10.3	34	
6	17.3			13.8			16.2			18	27		12.3		
7	20			21.6			19.6			19.4			16.5		
8							23			21					
9															
10															
residual deflection of spring mm	0.46	0.32	0.08	0.48	0.37	0.30	0.59	0.14	0.30	0.21	0.40	0.30	0.82	0.23	0.25

**Table 6 children-09-00107-t006:** Results for Co-Cr 1.2 mm arches.

Deflection mm	Co Cr 1.2 mm			Co Cr 1.2 mm			Co Cr 1.2 mm			Co Cr 1.2 mm			Co Cr 1.2 mm		
				1 Month in Artificial Saliva			2 Months in Artificial Saliva			1 Month in Artificial Saliva + Cola			2 2 Months in Artificial Saliva + Cola		
	l_3_ 15 mm F(N)	l_2_ 10 mm F(N)	l_1_ 5 mm F(N)	l_3_ 15 mm F(N)	l_2_ 10 mm F(N)	l_1_ 5 mm F(N)	l_3_ 15 mm F(N)	l_2_ 10 mm F(N)	l_1_ 5 mm F(N)	l_1_ 15 mm F(N)	l_2_ 10 mm F(N)	l_1_ 5 mm F(N)	l_3_ 15 mm F(N)	l_2_ de 10 mm F(N)	l_1_ 5 mm F(N)
1	4.6	6.9	31	3.3	7.8	17	5.2	6	9.8	4.6	5	12.2	3.6	2.3	6.9
2	12.4	20	61	7.9	16	36	9.7	10.5	22	8.8	12	37	6.1	6.3	23
3	19	29.8	85	11.9	23		16	17.5	43	17	24	48	12.4	13	38
4	23	31		15			24	24		22	34		20.5	24	
5	25			17			27	29		25			23.4	53	
6	25			25			27.5			26			24.5		
7	24									26.2					
8	23									25.5					
9															
10															
residual deflection of spring mm	3.44	0.65	0.7	1.45	0.33	0.33	2.58	0.86	0.20	3.15	0.70	0.45	2.62	0.90	0.33

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
