# Peer review of "Experimental Study on the Mechanical Behavior of Orthodontic Arches Exposed to the Environment in the Oral Cavity"

_children, 2022, doi:10.3390/children9010107_

Round 1

Reviewer 1 Report

Generally, the paper is informative to ortho clinicians. But, the study is not new or emerging. 

Author should add more references about similar study like those on mechanical behavior of orthodontics arches in various environment, otherwise there may be a knowledge gap for those readers who are not orthodontist or clinical researchers in this field.

Please replace Fig 2 with a new one with higher resolution, which will be helpful to readers, in that booth spring and cylinder are not clear enough to see details.

Ni Ti arch was widely used in initial stage of aligning in orthodontics treatment. If author can also provide results of bending Ni Ti it would be more informative.

Author Response

Dear reviewer,

Author should add more references about similar study like those on mechanical behavior of orthodontics arches in various environment, otherwise there may be a knowledge gap for those readers who are not orthodontist or clinical researchers in this field ---- thank you for the suggestion, i added on page 2 paragraph 3, 5 and 6.

Please replace Fig 2 with a new one with higher resolution, which will be helpful to readers, in that booth spring and cylinder are not clear enough to see details ---- i retake and changed the picture.

Ni Ti arch was widely used in initial stage of aligning in orthodontics treatment. If author can also provide results of bending Ni Ti it would be more informative ---- in page 2 , paragraph 5 i described that NiCr and CoCr are more stabel and are used in appliances that stay longer in oral cavity. So in my study i prolonged the exposure time to the enviroment (sa and sa+cola).  NiTi springs are more sensityve to the enviroment, but they are more often changed by the practitioner.

Thank you!

    Kind regards,

      Alexandru Zalana

Reviewer 2 Report

This manuscript is bad written. 

There are lot of grammar and style mistakes.

There is no any statistical methods described.

The authors put Tables and Graphs what what statistical method they used???

How they get all concusions? Based on what? 

Author Response

Dear reviewer,

This manuscript is bad written.  --- thank you, I tryed to corect my manuscript. Hope is better. Made several changes around the text. I used change tracking.

There is no any statistical methods described. ---- Thank you, I elaborated more on working method:page 3 paragraph 3 and page 6, paragraph 1 and 2.

The authors put Tables and Graphs what what statistical method they used??? ----- I remaked all tables and Graphs, translated them and tried to explain better.

How they get all concusions? Based on what? ---- Thank you, I tryed to explain more the results: page 16, paragraph 1; I also added more discutions: page 16 paragraph 3; page 17 paragraph 4,5 and 6; page 18 paragraph 1 and 2.

Thank you!

Kind regards, 

   Alexandru Zalana

Reviewer 3 Report

Dear authors,

The topic of your manuscript is very interesting. However, there are a lot of things that must be improved before its publication.

The introduction and Discussion section must be extended, especially the discussion section. You have to base more on the literature background.

Table descriptions should be corrected-it must go as follows: Table 1.....

You have to correct the citation. In the introduction section, the first citation starts with 25 9where is 1 and 2 and etc... The funny thing is that the references list ends with 24. It is chaos.

Conclusions-please remove the numbering

The references are written not in the rules of the journal

Author Response

Dear reviewer,

 Thank you!

The introduction and Discussion section must be extended, especially the discussion section. You have to base more on the literature background. ----- I added more paragraph in introduction:  page 2 paragraph 3, 5 and 6; I also added more discussions: page 16 paragraph 3; page 17 paragraph 4,5,6 and 7; page 18 paragraph 1 and 2. Added to conclusion: page 18 paragraph 9.

able descriptions should be corrected-it must go as follows: Table 1. ---- Thank you , I remaked all tables and pictures (graphs), translated them and tryed to better explain in working methods.

You have to correct the citation. In the introduction section, the first citation starts with 25 9where is 1 and 2 and etc... The funny thing is that the references list ends with 24. It is chaos. Conclusions-please remove the numbering ----- Thank you, I removed the numbering on citation.

The references are written not in the rules of the journal ----- I added more references, tried to respect the rules of journal, some of them are books.

Thank you!

Kind regards,

   Alexandru Zalana

Round 2

Reviewer 2 Report

Please add statystical methods paragraph.

Thank you

Author Response

Dear reviewer,

I added more descriptions. Page 3 paragraph 4, page 6 paragraph 4, page 7 picture 6 - Graph that show the linear regression for NiCr 0,7 mm. I also added refference 38 and 39 (Linear regression).

Thank you very much!

  Respectfully,

    Kind regards,

    Alexandru Stefan Zalana

Reviewer 3 Report

Dear authors,

The manuscript has been improved, however, there are still some things to change:

  1. You have to add citations in the Introduction and discussion section
  2. 2. The references style must be improved-it is still not correct. See the authors' guidelines at the website of the journal

Author Response

Dear reviewer,

I added citations: page 18 paragraph 1, page 17 paragraph 1 and 2.

I worked on references page 20 to 25

Thank you!

Respectfully,
    Kind regards,
    Alexandru Stefan Zalana